# Dental Pulp Stem Cells for Salivary Gland Regeneration—Where Are We Today?

**DOI:** 10.3390/ijms24108664

**Published:** 2023-05-12

**Authors:** David Muallah, Jan Matschke, Matthias Kappler, Lysann Michaela Kroschwald, Günter Lauer, Alexander W. Eckert

**Affiliations:** 1Department of Oral and Maxillofacial Surgery, University Hospital Hamburg-Eppendorf, Martinistraße 52, 20251 Hamburg, Germany; 2Department of Oral and Maxillofacial Surgery, Faculty of Medicine “Carl Gustav Carus”, Technische Universität Dresden, Fetscherstraße 74, 01307 Dresden, Germany; 3Department of Oral and Maxillofacial Plastic Surgery, Martin Luther University Halle-Wittenberg, 06120 Halle, Germany; 4Center for Translational Bone, Joint and Soft Tissue Research, University Hospital “Carl Gustav Carus”, Technische Universität Dresden, Fetscherstraße 74, 01307 Dresden, Germany; 5Department of Cranio Maxillofacial Surgery, Paracelsus Medical University, Breslauer Straße 201, 90471 Nuremberg, Germany

**Keywords:** dental pulp stem cells, DPSC, salivary glands, xerostomia, tissue engineering, differentiation, regenerative medicine, Sjogren syndrome

## Abstract

Xerostomia is the phenomenon of dry mouth and is mostly caused by hypofunction of the salivary glands. This hypofunction can be caused by tumors, head and neck irradiation, hormonal changes, inflammation or autoimmune disease such as Sjögren’s syndrome. It is associated with a tremendous decrease in health-related quality of life due to impairment of articulation, ingestion and oral immune defenses. Current treatment concepts mainly consist of saliva substitutes and parasympathomimetic drugs, but the outcome of these therapies is deficient. Regenerative medicine is a promising approach for the treatment of compromised tissue. For this purpose, stem cells can be utilized due to their ability to differentiate into various cell types. Dental pulp stem cells are adult stem cells that can be easily harvested from extracted teeth. They can form tissues of all three germ layers and are therefore becoming more and more popular for tissue engineering. Another potential benefit of these cells is their immunomodulatory effect. They suppress proinflammatory pathways of lymphocytes and could therefore probably be used for the treatment of chronic inflammation and autoimmune disease. These attributes make dental pulp stem cells an interesting tool for the regeneration of salivary glands and the treatment of xerostomia. Nevertheless, clinical studies are still missing. This review will highlight the current strategies for using dental pulp stem cells in the regeneration of salivary gland tissue.

## 1. Introduction

The salivary glands (SG) play an essential role in the integrity of the orofacial system as the production of saliva is crucial for digestion, articulation and oral immune defense. Different reasons can lead to xerostomia, the phenomenon of dry mouth. Most cases of xerostomia are caused by tumors, radiotherapy, hormonal changes or autoimmune diseases and lead to a tremendous decrease in quality of life [1,2,3,4,5,6,7,8,9]. Dryness of the mouth causes dysphagia, increased incidence of caries, impaired articulation and an imbalance of the oral microbiome [10,11]. Current therapy concepts are based on saliva substitutes (oral rinses, gels, powders and sprays) and systemic medication (e.g., pilocarpine, cevimeline). However, none of these approaches provides a satisfying outcome [12].

Recent studies show that tissue engineering is a promising approach for SG regeneration [13,14,15,16]. The principle of tissue engineering is to use cells in combination with different biomaterials and biochemical/physicochemical factors to build tissues in vitro [16,17,18,19,20]. Not only primary cells but also stem cells can be utilized for this purpose. By using embryonic stem cells (ESC), Tanaka et al. succeeded in engineering the first fully functional SG organoid in 2018 [13]. Nevertheless, harvesting ESCs is difficult as they must be extracted from the inner cell mass of the blastocyst. This can cause the destruction of the blastocyst and thereby jeopardizes the life of the embryo. Thus, ESCs raise ethical issues and cannot be implemented into clinical practice yet.

An alternative to ESCs is adult stem cells, which can be harvested from a tissue specimen of the patient. A popular source of adult stem cells is the adipose tissue [21]. Adipose tissue-derived stem cells (AdSC) were shown to have the ability of transdifferentiating to acinar cells in vitro [22]. Furthermore, they seem to have a protective effect on SG tissue undergoing irradiation, which may be caused by antioxidative features [23,24,25]. Nevertheless, AdSCs are harvested by liposuction, which is associated with several severe complications, such as bowel perforation, pneumothorax and sciatic nerve injury [26,27,28].

Besides AdSC, bone marrow-derived stem cells (BMdSC) were intensively investigated throughout recent years and showed similar effects on compromised SGs like AdSC [29,30,31]. However, harvesting of BMdSCs requires a biopsy of the bone marrow or the application of drugs like filgrastim, which can cause side effects such as bone pain or other musculoskeletal symptoms [32].

Another promising approach is the use of induced pluripotent stem cells (iPS), which are generated from reprogrammed somatic cells [33]. The conversion from somatic cells to iPS was first conducted by delivering ESC-specific genes via retroviruses [34]. Unfortunately, the tumorigenicity of iPS and the low efficacy of the conversion are only two of many challenges of this technique, which inhibit the implementation of iPS in the clinic.

To implement stem cell therapy in the clinic, easy and harmless accessibility to the cells is essential. For this reason, dental pulp stem cells (DPSC), which originate from the neural crest, have been the focus of regenerative medicine since their first description in 2000 [35,36,37,38]. DPSCs can be easily isolated from the pulp of extracted third molars in many different ways. One way is the incubation of cut teeth in a culture flask, as shown in Figure 1B. This method brings the risk of contamination with oral bacteria, which is why the extracted teeth should be preserved in the antibiotic medium for at least 12 h after extraction. The advantage of this method is the weighting of the pulp tissue, which ensures direct contact of the cells with the culture flask’s ground. This contact is necessary for the cells to attach to the ground and start migrating into the flask. To reduce the risk of contamination, the pulp can be peeled off the cracked tooth and put into the culture flask on its own (Figure 1C). This technique requires a higher level of experience, since the risk of the tissue floating away in the culture medium is higher.

The extraction of third molars is a routine intervention in maxillofacial surgery and can be conducted in a minimally invasive way and without general anesthesia. Most complications, such as swelling, pain and mild bleeding, are transient and resolve spontaneously within a few days. Severe complications are rare and can be avoided by selecting the right time and technique for extraction [39]. DPSCs can be cryopreserved, which makes it possible to store them and use them later on for autologous therapy when needed [40]. In contrast to other mesenchymal stem cells (MSC), DPSCs express transcription factors such as Oct-4, Sox2 and c-Myc, which are associated with pluripotency [41]. Still, they do not show tumor formation after transplantation as ESCs or induced pluripotent stem cells do [42]. Compared to BMdSCs, DPSCs show higher proliferation rates and a broader array of lineages [35,43]. They can be differentiated into tissues of all three germ sheets [44,45,46,47]. This extensive array of lineages makes DPSCs a precious tool for tissue engineering and regeneration of compromised SG tissue. Furthermore, DPSCs seem to have immunosuppressive effects by interfering with activated T-cells [48]. Therefore, they could be beneficial for the treatment of chronic inflammatory diseases such as rheumatoid arthritis, degenerative diseases of the nerval system, periodontitis or inflammatory bowel disease [49,50,51,52]. Similar to the aforementioned pathologies, Sjögren’s syndrome is also characterized by chronic inflammation. Hence, it could be hypothesized that patients suffering from Sjögrens’ syndrome would benefit from treatment with DPSC as well.

Taking these aspects into consideration, DPSCs may play an outstanding role in new approaches to regenerative medicine in the future [53]. Nevertheless, the use of DPSCs for the regeneration of SGs is still far away from clinical application. This review briefly describes the organogenesis of SGs to discuss frame conditions for regenerative approaches. An overview of the current literature and recent strategies for using DPSCs in regeneration of SGs will be provided. Subsequently, new aspects for further research will be discussed.

## 2. Organogenesis of Salivary Glands

The three major SGs are of different developmental origin. While the *Glandula submandibularis* and *Glandula sublingualis* derive from the endodermal germ sheet, the *Glandula parotis* originates from the ectoderm and therefore has the same origin as DPSCs [36,37,38]. As the first step of SG organogenesis, an epithelial placode comes to exist in the oral cavity during the seventh embryonic week. Subsequently, this placode infiltrates into the underlying mesodermal mesenchyme. Through dichotomous branching, a canalized system develops. The epithelial cells in the distal ends of the invaginating strands differentiate into acinar cells which produce primary saliva. During the morphogenesis the ectodermal cells continuously interact with the surrounding mesenchyme via several cytokines in both directions. Although this mechanism is not yet fully understood, FGF 10 was identified as one of these signaling molecules [44]. It is assumed that FGF 10 is expressed by cells of the mesenchyme and promotes the maturation of the epithelial gland tissue. But also vice versa, signals sent by the invaginating epithelium trigger the adjacent mesenchyme to differentiate into myoepithelial and stromal cells that surround the ducts and acini of the gland [54].

## 3. The Effect of DPSC on Primary Salivary Glands

The interaction between epithelial and mesenchymal cells led to different approaches to use DPSCs for SG regeneration. It could be shown that coculturing DPSCs with primary SG cells (SGC) on Matrigel™ increases the number and size of spontaneous acinus formation of the SGCs [55]. This in vitro observation by Reyes et al. was also confirmed by transplanting DPSCs and SGCs embedded in hyaluronic acid hydrogel subcutaneously into 2-month-old Rag1 null mice. Besides the typical acinar differentiation marker alpha amylase-1, other specific markers such as CD 44 and LAMP-1 were also increased compared to SGC-implantation alone [55]. It is hypothesized by many authors that DPSCs assume the role of embryonic mesenchyme that surrounds the invaginating epithelium during the organogenesis when they interact with primary acinar cells [55,56]. This assumption is even more strengthened by results observed when DPSCs were directly injected into compromised SGs of mice [57,58,59]. For instance, in a study by Yamamura et al., mice were exposed to irradiation to induce hyposalivation. Subsequently, DPSCs were injected into the submandibular glands. Eight weeks after irradiation, saliva flow was assessed. Mice treated with DPSCs showed a significantly higher saliva flow compared to the PBS control group [57]. Similar findings were reported for diabetic wistar rats by Narmada et al. and Suciadi et al. [58,59]. In summary, these approaches seem to prove the ability of DPSC to act as growth-supporting mesenchyme for acinar cells and thereby support the regeneration of compromised salivary glands.

## 4. The Immunomodulatory Effect of DPSC

BMdSCss, AdSC and umbilical cord-derived stem cells are known for having high immunomodulatory capacities as they are able to control inflammatory conditions [60,61,62,63,64,65]. It is believed that the immune response is regulated via cell–cell contact and/or paracrine production of soluble factors [66,67]. Some of these cells have even already reached phase I/II human trials [16,68,69].

According to DPSCs, only a few clinical trials have been conducted thus far and none of them aimed at the therapy of hyposalivation [70,71,72,73,74,75]. Nevertheless, DPSCs were reported to have immunomodulatory effects that even surpass those of other MSCs, which could be a precious tool in SG regeneration [76,77,78]. For instance, Ogata et al. found DPSCs to significantly surpass mesenchymal stem cells derived from bone marrow in a Sjögren’s syndrome mouse model according to anti-inflammatory factors such as IL-10, the downregulation of T-helper 17 cells and the upregulation of regulatory T cells [76,78]. Similarly, Du et al. injected DPSCs into the tail vein of mice with induced Sjögren’s syndrome [79]. The cells in this experiment were harvested from the pulp of exfoliated deciduous teeth, also known as stem cells from human exfoliated deciduous teeth (SHED). SHED were first isolated in 2003 and show similar characteristics as DPSCs [43]. It is noteworthy that they have an even higher proliferation rate when compared to DPSCs [80,81]. Du et al. reported that SHED have an anti-inflammatory and function-improving effect on damaged SGs of mice by migrating to the spleen and liver. The authors assume that pulp stem cells affect the SGs in an immunomodulatory way by influencing T-cell differentiation in these organs [79]. This assumption is substantiated by another study published in 2019, which provides evidence of the pulp stem cells’ effect on T-cells [77]. Ji found a decreased differentiation of CD4+ T-cells into T-helper 17 cells and, subsequently, a decreased secretion of IL-17 and TNF-α after coculturing DPSCs with peripheral blood mononuclear cells. Furthermore, the DPSCs promoted the polarization of CD4+ T-cells into regulatory T-cells, which have immunosuppressive effects [77]. Rasha et al. observed increased salivary flow rates and a reduction of oxidative stress after injecting DPSCs into the tail veins of diabetic rats [82]. Nevertheless, these experimental settings do not reveal whether the cell–cell communication between DPSCs and immune cells happens in a paracrine or juxtacrine way as reported for other MSCs. While the juxtacrine communication would require direct cell contacts, paracrine communication could be carried out by proteins secreted by the DPSCs. This would raise the question of whether the immunomodulatory effect of DPSCs could also be provided by using only supernatants of DPSC cultures.

This approach was investigated by Takeuchi et al. [83]. Instead of using DPSCs, they injected conditioned supernatant of a DPSC culture intravenously. Mice with an induced defect of the *Glandula submandibularis* subsequently showed an increased regeneration of the SGs compared to the control group [83]. Nevertheless, conditioned supernatants contain several substances that are redundant and have no benefit for the aspired purpose such as antibiotics, fungicides, fetal bovine serum and HEPES. While irresponsible use of antibiotics can lead to bacterial resistance, other substances are discussed as being toxic or allergenic [84,85]. Therefore, exosomes became an object of interest for many researchers. Exosomes are small vesicles containing peptides and nucleic acids that are produced by cells for intercellular communication. These vesicles can be derived from conditioned cell supernatant via centrifugation. It has been found that MSC-derived exosomes could suppress T-cell activation and thereby stabilize an immune homeostasis [86]. BMdSC-derived exosomes were successfully used to save salivary glands from diabetic complications in rats [30]. Also, exosomes of DPSCs were shown to have the capacity of suppressing inflammation via facilitation of macrophages [87]. Compared to BMdSC-derived exosomes, their immunosuppressing effects are even higher [77]. Unfortunately, the reason for this remains unclear. It may be assumed that DPSC-derived exosomes contain different compositions of proteins, lipids, cytokines and RNAs that are responsible for their superior immunomodulatory effect. Thus, further investigations are necessary before they can be applied in the clinic.

However, these studies further aggravate the hypothesis of DPSCs supporting the regeneration of compromised salivary glands. In the future, one of many possible applications could be the injection of DPSC exosomes into the SGs of patients suffering from Sjögren’s syndrome, which is associated with an infiltration of lymphocytes into the glands [78]. Moreover, other medical disciplines (e.g., rheumatology, plastic surgery, etc.) could benefit from immunomodulatory features of DPSC as well.

## 5. Differentiation of DPSCs into Acinar-like Cells

Nevertheless, the trials mentioned so far leave it unclear if DPSCs themselves could be differentiated into acinar cells to replace damaged SG tissue. A recent study by Yan et al. could clarify this question [15]. They induced the differentiation of DPSCs into acinar cells by performing coculture with primary cells of the submandibular gland. The cells were physically separated by a membrane that allowed the exchange of cytokines and other molecules but not the juxtacrine communication [15]. After 2 weeks, specific acinar markers such as amylase and cytokeratin 8 were observed in the DPSCs. They also shaped cobblestone-like islands, which are typical for acinar cells [15]. As a control group, fibroblasts were cocultured with acinar cells. The fibroblasts did not differentiate into acinar cells, which proves that the differentiation is an exclusive feature of the DPSCs. With this experimental setting, the authors could show that DPSCs not only support primary cells in growth and regeneration but that they can also be affected by primary cells and form SG-like tissue. Nevertheless, a detailed analysis of the cell signaling at the molecular level that induces the differentiation is still missing. Therefore, further experiments should be conducted to perform a protein analysis of the coculture’s supernatants at different time points to retrace proteomic changes during the induction process.

While typical monolayers of cells as used by Yan et al. are two-dimensional arrangements, in situ cells are organized three-dimensionally [15]. Therefore, three-dimensional cell cultures, such as spheroids, more closely resemble in vivo conditions. These spheroids are usually produced by seeding cells on low-cell adhesion plates. A new way of producing DPSC spheroids was introduced by Adine et al. using a special 3D bioprinting technology [14]. To generate the spheroids, cells were incubated with a solution containing gold and iron oxide and subsequently printed using magnets beneath the well plate. The cells on the spheroids’ surfaces could then be differentiated into SG-like cells using FGF 10. Furthermore, epithelial, ductal, myoepithelial and neural elements were detected in the spheroids after immunostaining. The organoids even produced α-amylase. Since the secretion of saliva is regulated by the autonomic nervous system, Adine et al. tried to stimulate the organoids with the neurotransmitter derivatives carbachol and isoproterenol. The cells reacted with intracellular calcium mobilization and a shift in transepithelial resistance, which suggests physiological integrity by an action potential [14]. In an ex vivo SG mouse model, the printed organoids even rescued epithelial growth after irradiation with a single dose of 7 Gy. Furthermore, the neural compartments of the ex vivo glands were integrated into the spheroids [88]. Nevertheless, an ex vivo model does not consider the potential immune response after the implantation of a graft. Still, this experiment shows the usefulness of 3D cultures for SG regeneration after irradiation.

Besides FGF 10, FGF 7 was also reported to induce acinar differentiation [89]. Akashi et al. reported an upregulated expression of acinar-specific markers such as aqua-porin 5 after treating DPSC with FGF 7 in vitro and in vivo [89]. During the organogenesis of SGs, an increased expression of FGF 7 and FGF 10 was observed in the embryonic mesenchyme surrounding the SG rudiments, which could explain the findings of these studies [13]. In conclusion, FGF 10 and FGF 7 seem to be potential additives for SG tissue engineering.

## 6. Discussion

DPSCs offer new aspects for SG tissue regeneration. They are easy to harvest and can be cryopreserved without losing their differentiation potential [90]. Compared to other stem cells, DPSCs have a higher proliferation rate, a broader array of lineages and a smaller risk of tumor formation [35,42,44,45,46,47]. Several studies have shown therapeutical effects of DPSCs, such as the differentiation to pancreatic tissue in rats with diabetes, anti-inflammatory effects in mice with rheumatoid arthritis, lupus erythematosus or COPD and even improved vascular function in patients with erectile dysfunction [49,50,51,72,91,92,93,94]. They either act by affecting the immune system or the primary cells or by differentiating into other cell types. Also, for SG tissue engineering, DPSCs hold application potential due to their ability to differentiate into acinar cells [14,15]. For this purpose, they could be embedded into a scaffold to replace damaged gland tissue in terms of a graft. Besides the transdifferentiation, DPSCs are able to affect acinar cells or immune cells after injection into compromised tissue or intravenously [58,59,79]. Even the supernatants of DPSC cultures offer useful effects on primary acinar cells as well as the immune system [78,83]. While the injection of DPSCs or their supernatants is technically easy, the transplantation of tissue-engineered grafts poses the challenge of nerval and vascular supply. While thin grafts could probably be supplied with nutrients and oxygen via diffusion from the adjacent tissues, for thicker grafts, which would accomplish a rather satisfying production of saliva, alternative steps such as microvascular surgery are necessary. Autologous submandibular gland transplantation as a therapy for *keratoconjunctivitis sicca* reveals some interesting aspects here. For this therapy, the whole submandibular gland is denervated and transplanted to the temporal region. While blood supply can directly be restored by microvascular surgery, a nerval connection cannot be implemented surgically. Still, the transplanted glands start to produce saliva after a few months and thereby improve the moistening of the eyes, which means that the glands become reinnervated after a hypofunctional period. The reinnervation was histologically proven in a rabbit model [95]. Zhang et al. hypothesized that the autonomic reinnervation originates from the auriculotemporal nerve, which runs through the temporal region close to the transplant, but also from the sympathetic plexus around the supplying arteries. According to these findings, tissue-engineered SG grafts should be transplanted close to a bigger nerve or ganglion (e.g., ganglion submandibulare or the lingual nerve) to increase the chance of innervation. Another approach could be to relocate a blood vessel (e.g., the facial artery) so that it runs directly through the graft. This may not only promote the reinnervation as shown by Zhang et al. but also the blood supply of the graft.

Besides cell differentiation, the composition of the scaffold/extracellular matrix is crucial for successful tissue-engineered grafts. None of the abovementioned studies investigated whether differentiated DPSCs stay acinar cells for long, which, indeed, depends on the surrounding extracellular matrix. Cells tend to dedifferentiate if they are not embedded in a supportive environment that matches their requirements. Thus, the perfect matrix for SG grafts needs to be functionalized with signaling molecules or growth factors such as FGF 10 or FGF 7, which were both shown to induce acinar differentiation in DPSCs and seem to play a key role in SG organogenesis [13,14,89]. Moreover, the optimal scaffold material for SG grafts needs to have mechanical properties that allow simple handling during surgery and that withstand the motions caused by speaking or eating after implantation of the graft. Furthermore, they should be degradable to be replaced by autologous tissue over time. Possible options are platelet-rich fibrin, hydrogels, collagen matrices or silk as they are already proved for tissue engineering in several studies and match the abovementioned criteria [96,97,98,99,100,101].

To sum up, although many in vitro assays and animal studies have already proven the value of DPSCs in SG regeneration, clinical studies are still missing (Table 1). For tissue engineering of SGs, useful materials for artificial extracellular matrices need to be studied. Nevertheless, DPSCs seem to be an outstanding tool for SG tissue regeneration since they are easy to harvest, suppress pathological immune reactions, regenerate compromised gland tissue and can be differentiated into the acinar cell lineage (Figure 2).

## Figures and Tables

**Figure 1 ijms-24-08664-f001:**
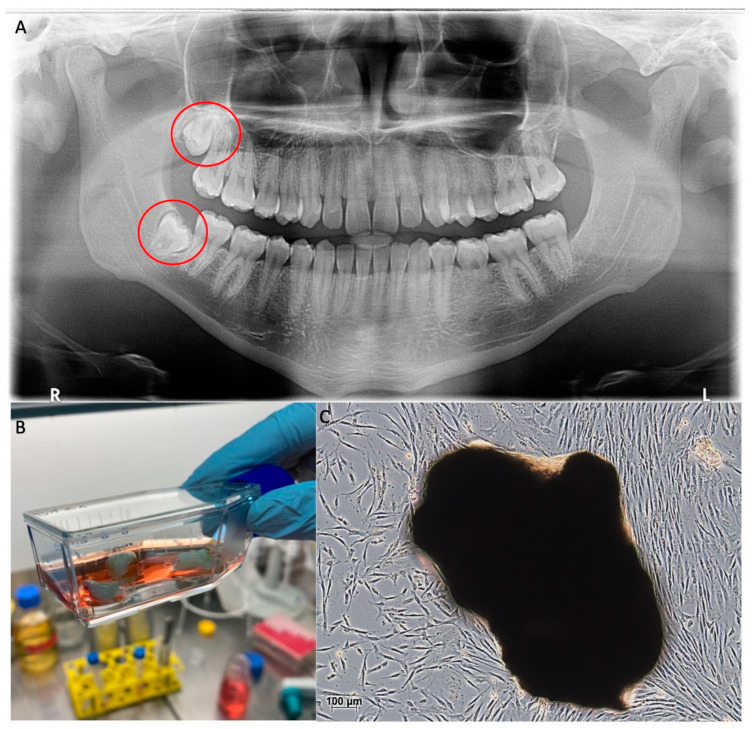
Harvesting of DPSCs; (**A**): Third molars in situ marked with red circles (panoramic X-ray); (**B**): Extracted third molars, split and digested in collagenase for explant culture; (**C**): DPSCs emigrating from dissolved pulp tissue (dark mass) and adhering to the culture flask investigated by light microscopy. This figure belongs to David Muallah, Department of Oral and Maxillofacial Surgery, University Hospital Hamburg-Eppendorf.

**Figure 2 ijms-24-08664-f002:**
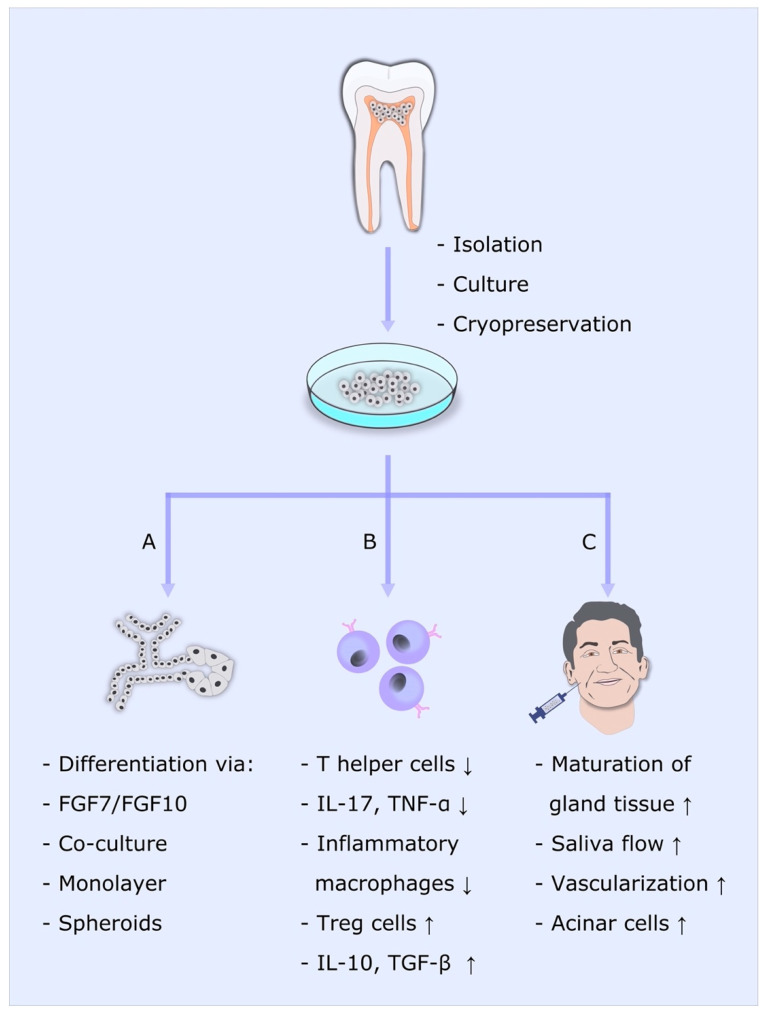
DPSCs for salivary gland regeneration. DPSCs can be isolated from the dental pulp, cultivated in vitro and cryopreserved. A: DPSCs can be differentiated toward acinar lineage by coculture or induction by FGF7/FGF10 in monolayers or as spheroids. B: DPSCs and DPSC-conditioned medium directly affect immune cells in various ways. C: DPSCs serve as supportive mesenchymal tissue after direct injection into salivary glands. This figure belongs to David Muallah, Department of Oral and Maxillofacial Surgery, University Hospital Hamburg-Eppendorf.

**Table 1 ijms-24-08664-t001:** Different mechanisms of DPSCs affecting salivary gland regeneration.

Mechanism	Author, Year
Effect on primary cells	
Increased development of acinar structures and expression of LAMP-1 and CD44 after coculture of human salivary gland cells with DPSCs	[55] Reyes et al., 2013
Increased saliva flow after DPSC injection into radiated salivary glands of mice	[57] Yamamura et al., 2013
Decreased acinar cell vacuolization and increased IL-10 serum levels after DPSC injection into diabetic rats’ salivary glands	[58] Narmada et al., 2019
Increase of vascularization, TGF-β serum level and acinar cell number after DPSC injection into diabetic rats’ salivary glands	[59] Suciadi et al., 2019
Immunomodulatory effects	
Decreased apoptotic cell number in salivary glands of diabetic rats after injection of DPSCs in tail veins; also reduced expression of ATG5 and Beclin-1 as well as suppression of Th1 and Tfh cells in spleen while increased number of Treg cells	[79] Du et al., 2019
Inhibition of CD4+T cells’ differentiation into T helper 17 cells and reduction of IL-17 and TNF-α, promotion of Treg cells and increased release of IL-10 and TGF-β	[77] Ji et al., 2019
Downregulation of caspase-3 and upregulation of VEGF, decreased blood glucose, improved gland weight and salivary flow in diabetic rats after injection of DPSCs into the tail vein	[82] Al-Serwi et al., 2021
Treatment of mice in salivary gland duct ligation model with DPSC-conditioned medium leads to increased expression of CK5, AQP5	[83] Takeuchi et al., 2020
DPSC exosomes caused macrophages to transform from proinflammatory phenotype to anti-inflammatory phenotype	[87] Shen et al., 2020
Differentiation of DPSC to acinar cells	
Differentiation of DPSC via coculture with acinar cells in monolayer and expression of specific acinar morphology and markers such as CK8, amylase	[15] Yan et al., 2020
3D culture of DPSCs differentiated into acinar-like cells using FGF 10	[14] Adine et al., 2018
Differentiation of DPSC into acinar-like cells expressing AQP5 and αSMA after induction via FGF 7	[79] Akashi et al., 2021

## Data Availability

The data used for this review can be found on Pubmed data base and Google Scholar.

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
