# Peer review of "Dental Pulp Stem Cells for Salivary Gland Regeneration—Where Are We Today?"

_ijms, 2023, doi:10.3390/ijms24108664_

Round 1
Reviewer 1 Report
Dear authors.
The paper submitted for review is interesting and deals with the important topic of the use of dental pulp stem cells in regenerative medicine.
1. After reading the paper, please provide in table form the proposed mechanisms of action and references to the literature
2 Make a diagram - graphic showing the described mechanisms of action.
3. Extracting pulp from a cut tooth as in the photograph (placed in culture) as in Fig. 1 is inefficient. The next photo shows a culture from an explant. Please also refer to the methodology for the isolation of tooth pulp in a more reproducible manner. The presented method with tooth fragments is problematic and risky.
Author Response
Dear Editor,
Many thanks to the reviewers for reading our manuscript and giving such helpful comments. We have revised our article according to the suggestions of the reviewers in the form of a major revision. We hope that we now address the concerns of the reviewer in an adequate manner.
Reviewer 1
Criticism 1 - After reading the paper, please provide in table form the proposed mechanisms of action and references to the literature.
Regarding the suggestion of reviewer 1, we included a table (attached as table 1) with the proposed mechanisms.
Criticism 2 - Make a diagram - graphic showing the described mechanisms of action.
We have replaced Figure 2 in the new version of the manuscript with a more attractive and informative figure, hoping to match reviewer’s 1 suggestions and to provide the reader with an even better overview through this figure.
Criticism 3 - Extracting pulp from a cut tooth as in the photograph (placed in culture) as in Fig. 1 is inefficient. The next photo shows a culture from an explant. Please also refer to the methodology for the isolation of tooth pulp in a more reproducible manner. The presented method with tooth fragments is problematic and risky.
We agree that the extraction of DPSC out of a cut tooth is on high risk of contamination with oral bacteria. Here, the use of broad-spectrum antibiotics should be carried out to support this technique, as mentioned in the manuscript. Nevertheless, the advantage of this method is the fixation of pulp tissue on the ground of the culture flask, which is necessary for the cells to attach to the ground. Leaving only the pulp tissue in the flask without weighting it brings the risk of the tissue floating away in the medium, which makes the attachment to the ground impossible. There are several ways to isolate DPSC each having advantages and disadvantages. However, in this review we did not try to focus on the actual way of isolation but on the value for salivary gland regeneration. The photographs serve to give the reader an idea of how the isolation can work. Nevertheless, we have now shortly addressed this issue in the new version of the paper (introduction part, line 75-86).

Reviewer 2 Report
This review focuses on salivary gland regeneration with DPSCs.
Many articles have been reported on the therapeutic effects of DPSCs and DPSC culture supernatants on salivary gland damage.
Whether these therapeutic mechanisms protect or regenerate salivary glands should be clearly described.
Line70. For stem cell therapy to be implemented in clinical practice, it is essential that the cells be easily accessible and harmless; there have been reports of salivary gland regeneration studies using iPS cells and peripheral blood stem cells.
Figure 1C. There is no comment on the black mass in the center of the figure. Is it a sphere or dissolved tissue?
Paragraph 4. describes the function of Immunomodulator in DPSC; it is better to separate the tissue engineering aspect of DPSC from the immunomodulatory aspect. There are many papers on stem cell immunomodulation and the author should describe the characteristics of DPSCs in comparison to other stem cells. Further paper searches and additions to the bibliography are needed; the immunosuppressive function of DPSC-CM and SHED-CM has already been reported in other papers. That paper should be mentioned.
Line 138. although there is a review of DPSC, there is no description of clinical trials of DPSC. Why describe only the clinical trials of AdSC and BMdSC?
Line164. add mention to Takeuchi et al.
Line168. several substances in the culture supernatant should be specifically listed and discussed.
Line176. should discuss the differences in composition of DPSC-Exo compared to BMdSC-Exo.
Line182. therapeutic effects of DPSC have already been reported for many diseases. This point should also be discussed.
Lines 234-236: Here, the functional markers of acinar cells are discussed. I do not understand the continuity between the preceding and following sentences. The need for this discussion should be more clear.
Figure 2. It does not seem necessary to explain this treatment protocol in a diagram.  It would be more meaningful to have a table that summarizes how the DPSC regenerates the salivary glands.
Author Response
Dear Editor,
Many thanks to the reviewers for reading our manuscript and giving such helpful comments. We have revised our article according to the suggestions of the reviewers in the form of a major revision. We hope that we now address the concerns of the reviewer in an adequate manner.
Reviewer 2
Criticism 1 - Many articles have been reported on the therapeutic effects of DPSCs and DPSC culture supernatants on salivary gland damage. Whether these therapeutic mechanisms protect or regenerate salivary glands should be clearly described.
The cited studies suggest that DPSC support the regeneration of compromised salivary glands in different ways. We integrated this aspect in the respective sections of the new manuscript (line 138-144, line 194-195).
Criticism 2 - Line70. For stem cell therapy to be implemented in clinical practice, it is essential that the cells be easily accessible and harmless; there have been reports of salivary gland regeneration studies using iPS cells and peripheral blood stem cells.
We agree, that iPS cells made of blood stem cells or other somatic cells are a promising approach for tissue engineering. Nevertheless, from our point of view this technique is still in a very experimental stage. It is very expansive and brings the risk of tumorgenicity. However, we integrated this aspect into the introduction of our revised manuscript (line 68-72).
Criticism 3 - Figure 1C. There is no comment on the black mass in the center of the figure. Is it a sphere or dissolved tissue?
We clarified this aspect in the caption of figure 1. We added: “ C: DPSC emigrating from dissolved pulp tissue (dark mass) and adhering to the culture flask investigated by light microscopy”.
Criticism 4 - Paragraph 4. describes the function of Immunomodulator in DPSC; it is better to separate the tissue engineering aspect of DPSC from the immunomodulatory aspect. There are many papers on stem cell immunomodulation and the author should describe the characteristics of DPSCs in comparison to other stem cells. Further paper searches and additions to the bibliography are needed; the immunosuppressive function of DPSC-CM and SHED-CM has already been reported in other papers. That paper should be mentioned.
We separated the tissue engineering aspect (paragraph 5) from the immunomodulatory aspect (paragraph 4), as suggested by reviewer 2. We also included a comparison of the DPSCs’ immunomodulatory effect with bone marrow derived stem cells (line 154-156, line 188-189). From many studies discussing the immunomodulatory effect of stem cells, we included 18 publications, that were the most relevant from our point of view.
Criticism 5 - Line 138. although there is a review of DPSC, there is no description of clinical trials of DPSC. Why describe only the clinical trials of AdSC and BMdSC?
To our knowledge there are no clinical trials of DPSC used for salivary gland regeneration yet. However, we added some references of clinical trials using DPSC in general (line 151-152).
Criticism 6 - Line164. add mention to Takeuchi et al.
Reference was added (line 176). Thanks for your suggestion.
Criticism 7 - Line168. several substances in the culture supernatant should be specifically listed and discussed.
Since there are several different protocols for DPSC medium, it is difficult to refer to every substance that could be included. We listed the most common substances and pointed out why they could be dangerous (line 180-182).
Criticism 8 - Line176. should discuss the differences in composition of DPSC-Exo compared to BMdSC-Exo.
Unfortunately, the composition of DPSC-Exo and BMdSC-Exo is not compared in recent studies. We assume that there are different compositions of RNA and cytokines, which cause the different effects on immune cells. We added this aspect to paragraph 4 (line 189-192).
Criticism 9 - Line182. therapeutic effects of DPSC have already been reported for many diseases. This point should also be discussed.
Some therapeutical effects in other diseases are now mentioned in the discussion part of the manuscript (line 245-249) as suggested by reviewer 2.
Criticism 10 - Lines 234-236: Here, the functional markers of acinar cells are discussed. I do not understand the continuity between the preceding and following sentences. The need for this discussion should be more clear.
We agree with reviewer 2, that discussing functional markers is not relevant here. We therefore removed this sentence (line 251).
Criticism 11 - Figure 2. It does not seem necessary to explain this treatment protocol in a diagram. It would be more meaningful to have a table that summarizes how the DPSC regenerates the salivary glands.
We added a table (table 1) as overview of the different ways DPSC could improve salivary gland regeneration. We also added a new figure, which is superior to the last one.
Round 2
Reviewer 1 Report
Dear authors
Thank you for the corrections made to your publication and the prepared answers to the questions. However, I do not see the table you write about in your answer to the first question. Please supplement it.
Author Response
Dear editor, Dear reviewers,
thank you for your helpful comments that have improved our manuscript.
Regarding Reviewer 1's comment on Table 1. This document was submitted separately as a single file and was therefore not part of the manuscript. However, we added this table 1 to the end of our file manuscript so that it won't get lost this time.
Thank you very much.

Reviewer 2 Report
Sufficient improvements have been made.
Author Response

(The authors gave the same response as above.)
